# Understanding Collections of Related Datasets Using Dependent MMD Coresets

**Sinead A. Williamson** [1,*] and **Jette Henderson** [2]

1 Department of Statistics and Data Science, University of Texas at Austin, Austin, TX 78712, USA
2 CognitiveScale, Austin, TX 78759, USA; jhenderson@cognitivescale.com
* Correspondence: sinead@austin.utexas.edu

**Abstract:** Understanding how two datasets differ can help us determine whether one dataset under-represents certain sub-populations, and provides insights into how well models will generalize across datasets. Representative points selected by a maximum mean discrepancy (MMD) coreset can provide interpretable summaries of a single dataset, but are not easily compared across datasets. In this paper, we introduce dependent MMD coresets, a data summarization method for collections of datasets that facilitates comparison of distributions. We show that dependent MMD coresets are useful for understanding multiple related datasets and understanding model generalization between such datasets.

**Keywords:** coresets; data summarization; maximum mean discrepancy; interpretability





## 1. Introduction

When working with large datasets, it is important to understand your data. If a dataset is not representative of your population of interest, and no appropriate correction is made, then models trained on this data may perform poorly in the wild. Sub-populations that are under-represented in the training data are likely to be poorly served by the resulting algorithm, leading to unanticipated or unfair outcomes—something that has been observed in numerous scenarios including medical diagnoses [1,2] and image classification [3,4].

In low-dimensional settings, it is common to summarize data using summary statistics such as marginal moments or label frequencies, or to visualize univariate or bivariate marginal distributions using histograms or scatter plots. As the dimensionality of our data increases, such summaries and visualizations become unwieldy, and ignore higher-order correlation structure. In structured data such as images, such summary statistics can be hard to interpret, and can exclude important information about the distribution [5,6]—the per-pixel mean and standard deviation of a collection of images tells us little about the overall distribution. Further, if our data are not labeled, or are only partially labeled, we cannot make use of label frequencies to assess class balance.

In such settings, we can instead choose to present a set of exemplars that capture the diversity of the data. This is particularly helpful for structured, high-dimensional data such as images or text, that can easily be qualitatively assessed by a person. A number of algorithms have been proposed to find such a set of exemplars [7–17]. Many of these algorithms can be seen as constructing a coreset for the dataset—a (potentially weighted) set of exemplars that behave similarly to the full dataset under a certain class of functions. In particular, coresets that minimize the maximum mean discrepancy [18] (MMD) between coreset and data have recently been used for understanding data distributions [11,13]. Further, evaluating models on such MMD-coresets have been shown to aid in understanding model performance [11].

In addition to summarizing a single dataset, we may also wish to compare and contrast multiple related datasets. For example, a company may be interested in characterizing differences and similarities between different markets. A machine learning practitioner

may wish to know whether their dataset is similar to that used to train a given model. A researcher may be interested in understanding trends in posts or images on social media. Here, summary statistics offer interpretable comparisons: we can plot the mean and standard deviation of a given marginal quantity over time, and easily see how it changes [19,20]. By contrast, coresets are harder to compare, since the exemplars selected for two datasets $X_1$ and $X_2$ will not in general overlap.

In this paper, we introduce dependent MMD coresets, a new tool for characterizing related datasets and understanding model behavior across such datasets. These dependent MMD coresets provide a low-dimensional summary of a collection of datasets, that allows easy comparison across datasets. A dependent MMD coreset for a collection of datasets constructs a collection of exemplars, that is shared across all datasets. Each dataset assigns a different weight vector to these exemplars, so that the weighted exemplars approximate the dataset. These weights allow us to easily see which exemplars are relevant to which datasets, and comparing two sets of weights provides a simple way of showing how the corresponding datasets differ.

The use of shared exemplars makes it easy to compare two or more datasets, by providing a common language. Consider comparing two datasets of faces. If we independently constructed representations of each dataset—for example, using two independent MMD coresets—we would obtain two disjoint sets of weighted exemplars. Visually assessing the similarity between two sets would involve considering both the similarities of the images and the similarities in the weights. Conversely, with a dependent MMD coreset, the exemplars would be shared between the two datasets. Similarity can be assessed by considering the relative weights assigned in the two marginal coresets. This in turn leads to easy summarization of the difference between the two datasets, by identifying exemplars that are highly representative of one dataset, but less representative of the other.

In addition to understanding the difference between multiple datasets, dependent MMD coresets allow us to qualitatively explore the behavior of algorithms on these datasets. The shared set of exemplars provides representative points at which to evaluate the algorithm. Looking at the relative weights of these exemplars in the different datasets paints a picture of the relative performances we would expect between those datasets. This is particularly useful when a model has been trained on one dataset, but we wish to apply it to a second dataset: looking at exemplars that are highly representative of the second dataset, but not the first, allows us to identify potential failure modes.

We begin by considering existing coreset methods for data and model understanding in Section 2, before discussing their limitations and proposing our dependent MMD coreset in Section 3. A greedy algorithm to select dependent MMD coresets is provided in Section 3.4. In Section 4, we evaluate the efficacy of this algorithm, and show how the resulting dependent coresets can be used for understanding collections of image datasets and probing the generalization behavior of machine learning algorithms. We summarize notation used in this paper in Table 1.

**Table 1.** Notation used in this paper.

| | |
|---|---|
| $\mathcal{T}$ | set that indexes datasets and associated measures |
| $X_t = (x_{t,1}, \ldots, x_{t,n_t}) \in \mathcal{X}^{n_t}$ | a dataset indexed by $t \in \mathcal{T}$ |
| $\mathbb{P}_t$ | true distribution at $t \in \mathcal{T}$, $X_t \sim \mathbb{P}_t$ |
| $U = (u_1, \ldots, u_{n_u})$ | set of candidate locations |
| $\delta_u$ | Dirac measure (i.e., point mass) at $u$. |
| $\mathbb{Q}_t$ | a probability measure used to approximate $\mathbf{P}_t$, that takes the form $\sum_{i \in S} w_{t,i} \delta_{u_i}$, where $S \subset [n_u]$ |

## 2. Background and Related Work

### 2.1. Coresets and Measure-Coresets

A coreset is a "small" summary of a dataset $X$, which can act as a proxy for the dataset under a certain class of functions $\mathcal{F}$. Concretely, a weighted set of points $\{(w_i, u_i)\}_{i \in S}$ are an $\epsilon$ strong coreset for a size-$n$ dataset $X$ with respect to $\mathcal{F}$ if

$$\left| \frac{1}{n} \sum_{i=1}^{n} f(x_i) - \frac{1}{|S|} \sum_{j \in S} w_j f(u_j) \right| \leq \epsilon$$

for all $f \in \mathcal{F}$ [21].

A measure coreset [22] generalizes this idea to assume that $X$ are independently and identically distributed samples from some distribution $\mathbb{P}$. A measure $\mathbb{Q}$ is an $\epsilon$-measure coreset for $\mathbb{P}$ with respect to some class $F$ of functions if

$$\sup_{f \in \mathcal{F}} |\mathbb{E}_{X \sim \mathbb{P}}[f(X)] - \mathbb{E}_{Y \sim \mathbb{Q}}[f(Y)]| \leq \epsilon. \tag{1}$$

The left hand side of Equation (1) describes an integral probability metric [23], a class of distances between probability measures parametrized by some class $\mathcal{F}$ of functions. Different choices of $\mathcal{F}$ yield different distributions (Table 2).

**Table 2.** Some examples of integral probability metrics.

| Distance | $\mathcal{F}$ |
|---|---|
| 1-Wasserstein distance | $\{f : \|\nabla f\|_1 \leq 1\}$ |
| Maximum mean discrepancy | $\{f : \|f\|_{\mathcal{H}} \leq 1\}$ for some RKHS $\mathcal{H}$ |
| Total variation | $\{f : \|f\|_\infty \leq 1\}$ |

### 2.2. MMD-Coresets

In this paper, we consider the case where $\mathcal{F}$ is the class of all functions that can be represented in the unit ball of some reproducing kernel Hilbert space (RKHS) $\mathcal{H}$—a very rich class of continuous functions on $\mathcal{X}$. This corresponds to a metric known as the maximum mean discrepancy [18] (MMD),

$$\text{MMD}(\mathbb{P}, \mathbb{Q}) = \sup_{f \in \mathcal{H}} |\mathbb{E}_{X \sim \mathbb{P}}[f(X)] - \mathbb{E}_{Y \sim \mathbb{Q}}[f(Y)]|. \tag{2}$$

An RKHS can be defined in terms of a mapping $\Phi : \mathcal{X} \rightarrow \mathcal{H}$, which in turn specifies a kernel function $k(x, x') = \langle \Phi(x), \Phi(x') \rangle_{\mathcal{H}}$. A distribution $\mathbb{P}$ can be represented in this space in terms of its mean embedding, $\mu_P = \mathbb{E}_{\mathbb{P}}[\Phi(x)]$. The MMD between two distributions equivalently can be expressed in terms of their mean embeddings, $\text{MMD}(\mathbb{P}, \mathbb{Q})^2 = \|\mu_P - \mu_Q\|_{\mathcal{H}}^2$,

An $\epsilon$-MMD coreset for a distribution $\mathbb{P}$ is a finite, atomic distribution $\mathbb{Q} = \sum_{i \in S} w_i \delta_{u_i}$ such that $\text{MMD}(\mathbb{P}, \mathbb{Q})^2 \leq \epsilon^2$. We will refer to the set $\{u_i\}_{i \in S}$ as the support of $\mathbb{Q}$, and refer to individual locations in the support of $\mathbb{Q}$ as exemplars.

In practice, we are unlikely to have access to $\mathbb{P}$ directly, but instead have samples $X := (x_1, \ldots, x_n) \sim \mathbb{P}$. If $\mathbb{Q} = \sum_{i \in S} w_i \delta_{u_i}$, we can estimate $\text{MMD}(\mathbb{P}, \mathbb{Q})^2$ as

$$\widehat{\text{MMD}^2}(X, \mathbb{Q}) = \frac{1}{n^2} \sum_{i=1}^{n} \sum_{j=1}^{n} k(x_i, x_j) + \sum_{i \in S} \sum_{j \in S} w_i w_j k(u_i, u_j) - \frac{2}{n} \sum_{i=1}^{n} \sum_{j \in S} w_j k(x_i, u_j).$$

We, therefore, define an $\epsilon$-MMD coreset for a dataset $X$ as a finite, atomic distribution $\mathbb{Q}$ such that $\widehat{\text{MMD}^2}(X, \mathbb{Q}) \leq \epsilon^2$—or equivalently, whose mean embedding $\mu_Q$ in $\mathcal{H}$ is close to the empirical mean embedding $\hat{\mu}_X$ so that $\|\mu_Q - \hat{\mu}_X\|_{\mathcal{H}}^2 \leq \epsilon^2$.

A number of algorithms have been proposed that correspond to finding $\epsilon$-MMD coresets, under certain restrictions on $\mathbb{Q}$ (While most of these algorithms do not explicitly use coreset terminology, the resulting set of samples, exemplars or prototypes meet the definition of an MMD coreset for some value of $\epsilon$). Many of these algorithms greedily construct an MMD coreset, adding exemplars one-by-one based on some criteria. For example, kernel herding [14,24,25] can be seen as finding an MMD coreset $\mathbb{Q}$ for a known distribution $\mathbb{P}$, with no restriction on the support of $\mathbb{Q}$. The greedy prototype-finding algorithm used by [11] can be seen as a version of kernel herding, where $\mathbb{P}$ is only observed via a set of samples $X$, and where the support of $\mathbb{Q}$ is restricted to be some subset of a collection of candidates $U$ (often chosen to be the data set $X$). Versions of this algorithm that assign weights to the atoms in $\mathbb{Q}$ are proposed in [13].

Other methods start from the full dataset, and repeatedly discard points to construct a coreset [15–17,26]. Loosely, these methods repeatedly partition the dataset based on a discrepancy criterion, and then discard one half of the partition. Compared with the greedy methods, these approaches typically obtain smaller coresets for a given $\epsilon$ [15,17]. As shown by [27], random sampling also provides a way to construct an MMD-coreset.

As in [11,13], in this paper we require the support of our coreset to be a subset of some finite set of candidates $U$, indexed by $1, \ldots, n_U$. In other words, our measure coresets will take the form $\mathbb{Q} = \sum_{i \in S} w_i \delta_{u_i}$, where $S \subset [n_U]$.

### 2.3. Coresets for Understanding Datasets and Models

The primary application of coresets is to create a compact representation of a large dataset, to allow for fast inference on downstream tasks (see [28] for a recent survey). However, such compact representations have also proved beneficial in interpretation of both models and datasets.

While humans are good at interpreting visual data [29], visualizing large quantities of data can become overwhelming due to the sheer quantity of information. Coresets can be used to filter such large datasets, while still retaining much of the relevant information.

The MMD-critic algorithm [11] uses a fixed-dimension, uniformly weighted MMD coreset, which they refer to as "prototypes", to summarize collections of images. Gurumoorthy et al. [13] extends this to use a weighted MMD coreset, showing that weighted prototypes allow us to better model the data distribution, leading to more interpretable summaries. Zheng et al. [30] show how unweighted MMD coresets can be used to represent spatial point processes such as spatial location of crimes.

Techniques such as coresets that produce representative points for a dataset can also be used to provide interpretations and explanations of the behavior of models on that dataset. Case-based reasoning approaches use representative points to describe "typical" behavior of a model [11,31–33]. Considering the model's output on such representative points can allow us to understand the model's behavior.

Viewing the model's behavior on a collection of "typical" points in our dataset also allows us to get an idea of the overall model performance on our data. Evaluating a model on a coreset can give an idea of how we expect it to perform on the entire dataset, and can help identify failure modes or subsets of the data where the model performs poorly.

### 2.4. Criticising MMD Coresets

While MMD coresets are good at summarizing a distribution, since the coreset is much smaller than the original dataset, there are likely to be outliers in the data distribution that are not well explained by the coreset. The MMD-critic algorithm supplements the "prototypes" associated with the MMD coreset with a set of "criticisms"—points that are poorly modeled by the coreset [11] .

Recall from Equation (5) that the MMD between two distributions $\mathbb{P}$ and $\mathbb{Q}$ corresponds to the maximum difference in the expected value on the two spaces, of a function

that can be represented in the unit ball of a Hilbert space $\mathcal{H}$. The function $f$ that achieves this maximum is known as the witness function, and is given by

$$f(x^*) = \mathbb{E}_{X \sim \mathbb{P}}[k(x^*, X)] - \mathbb{E}_{Y \sim \mathbb{Q}}[k(x^*, Y)].$$

When we only have access to $\mathbb{P}$ via a size-$n$ sample $X$, and where $\mathbb{Q} = \sum_{i \in S} w_i \delta_{u_i}$, we can approximate this as

$$\hat{f}(x^*) = \frac{1}{n} \sum_{i=1}^{n} k(x^*, X_i) - \sum_{j \in S} w_j k(x^*, u_j).$$

Criticisms of an MMD coreset for a data set $X$ are selected as the points in $X$ with the largest values of the witness function. Kim et al. [11] show that the combination of prototypes and criticisms allow us to visually understand large collections of images: the prototypes summarize the main structure of the dataset, while the criticisms allow us to represent the extrema of a distribution. Criticisms can also augment an MMD coreset in a case-based reasoning approach to model understanding, by allowing us to consider model behavior on both "typical" and "atypical" exemplars.

### 2.5. Dependent and Correlated Random Measures

Dependent random measures [34,35] are distributions over collections of countable measures $\mathbb{P}_t = \sum_{i=1}^{\infty} w_{t,i} \delta_{u_{t,i}}$, indexed by some set $\mathcal{T}$, such that the marginal distribution at each $t \in \mathcal{T}$ is described by a specific distribution. In most cases, this marginal distribution is a Dirichlet process, meaning that the $\mathbb{P}_t$ are probability distributions. Most dependent random measures either keep the weights $w_{t,i}$ or the atom locations $u_{t,i}$ constant accross $t$, to assist identifiability and interpretability.

In a Bayesian framework, dependent Dirichlet processes are often used as a prior for time-dependent mixture models. In settings where the atom locations (i.e., mixture components) are fixed but the weights vary, the posterior mixture components can be used to visualize and understand data drift [36,37]. The dependent coresets presented in this paper can be seen as deterministic, finite-dimensional analogues of these posterior dependent random measures.

## 3. Understanding Multiple Datasets Using Coresets

As we have seen, coresets provide a way of summarizing a single distribution. In this section, we discuss interpretational limitations that arise when we attempt to use coresets to summarize multiple related datasets (Section 3.1), before proposing dependent MMD coresets in Section 3.2 and discussing their uses in Section 3.3.

### 3.1. Understanding Multiple Datasets Using MMD-Coresets

If we have a collection $\{X_t\}_{t \in \mathcal{T}}$ of datasets, we might wish to find $\epsilon$-MMD coresets for each of the $X_t$, in the hope of not just summarizing the individual datasets, but also of easily comparing them. However, if we want to understand the relationships between the datasets, in addition to their marginal distributions, comparing such coresets in an interpretable manner is challenging.

An MMD coreset selects a set $\{u_i\}_{i \in S}$ of points from some set of candidates $U$. Even if two datasets $X$ and $Y$ are sampled from the same underlying distribution (i.e., $X, Y \overset{\text{iid}}{\sim} \mathbb{P}$), and the set $U$ of available candidates is shared, the optimal MMD-coreset for the two datasets will differ in general. Sampling error between the two distributions means that $\widehat{\text{MMD}^2}(X, \mathbb{Q}) \neq \widehat{\text{MMD}^2}(Y, \mathbb{Q})$ for any candidate coreset $Q$ unless $X \equiv Y$, and so the optimal coreset will typically differ between the two datasets.

Figure 1 shows that, even if two distributions $X$ and $Y$ are sampled from the same underlying distribution, and their coreset locations are selected from the same collection $U$, the two coresets will not be identical. Here, we see two datasets (Figure 1b,c) generated from the same mixture of three equally weighted Gaussians (Figure 1a). Below (Figure 1d,e),

we have selected a coreset for each dataset (using the algorithm that will be introduced in Section 3.4), with locations selected from a shared set $U$. While the associated coresets are visually similar, they are not the same.

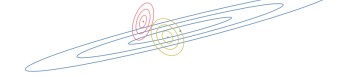

(**a**) A mixture of three equally weighted Gaussians.

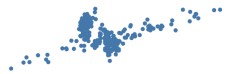

(**b**) First sample of 250 observations from the distribution in Figure 1a.

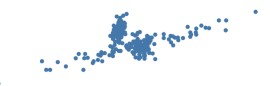

(**c**) Second sample of 250 observations from the distribution in Figure 1a.

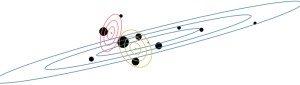

(**d**) MMD coreset for the sample in Figure 1b.

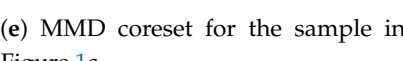

(**e**) MMD coreset for the sample in Figure 1c.

**Figure 1.** (**a**) Three equally weighted Gaussians (lines show 1, 2, 3 standard deviations of each component). (**b**,**c**) Independently sampled datasets from the mixture of three Gaussians. (**d**,**e**) MMD coresets for the three-Gaussian datasets.

This is magnified if we look at a high-dimensional dataset. Here, the relative sparsity of data points (and candidate points) in the space means that individual locations in $\mathbb{Q}_X$ might not have close neighbors in $\mathbb{Q}_Y$, even if $X$ and $Y$ are sampled from the same distribution. Further, in high dimensional spaces, it is harder to visually assess the distance between two exemplars. These observations make it hard to compare two coresets, and gain insights about similarities and differences between the associated datasets.

To demonstrate this, we constructed two datasets, each containing 250 randomly selected, female-identified US highschool yearbook photos from the 1990s. Figure 2 shows MMD-coresets obtained for the two datasets (See Section 4.2.1 for full details of dataset and coreset generation.) While both datasets were selected from the same distribution, there is no overlap in the support of the two coresets. Visually, it is hard to tell that these two coresets are representing samples from the same distribution.

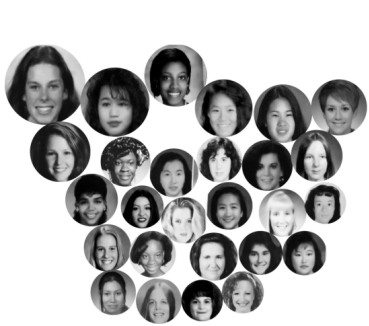

(**a**) MMD coreset for a set of 250 randomly selected yearbook photos from the 1990s.

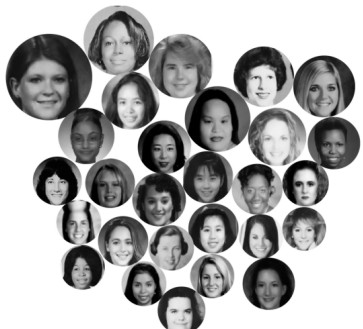

(**b**) MMD coreset for a second set of 250 randomly selected yearbook photos from the 1990s.

**Figure 2.** Independently learned MMD coresets for two randomly selected dataset of 250 female-identified photographs from US yearbooks in the 1990s. Area of each bubble is proportional to weight of the corresponding exemplar.

### 3.2. Dependent MMD Coresets

The coresets in Figure 2 are hard to compare due to their disjoint supports (i.e., the fact that there are no shared exemplars). Comparing the two coresets involves comparing multiple individual photos and assessing their similarities, in addition to incorporating the information encoded in the associated weights. To avoid the lack of interpretability resulting from dissimilar supports, we introduce the notion of a dependent MMD coreset.

Given a collection of datasets $\{X_t\}_{t \in \mathcal{T}}$, the collection of finite, atomic measures $\{\mathbb{Q}_t\}_{t \in \mathcal{T}}$, is an $\epsilon$-dependent MMD coreset if

$$\widehat{\text{MMD}^2}(X_t, \mathbb{Q}_t) \leq \epsilon^2 \tag{3}$$

for all $t \in \mathcal{T}$, and if the $\mathbb{Q}_t$ have common support, i.e.,

$$\mathbb{Q}_t = \sum_{i \in S} w_{t,i} \delta_{u_i}, \tag{4}$$

where $\{u_i\}_{i \in S}$ is a subset of some candidate set $U$.

In Equation (4), the exemplars $u_i$ are shared between all $t \in \mathcal{T}$, but the weights $w_{t,i}$ associated with these exemplars can vary with $t$. Taking the view from Hilbert space, we are restricting the mean embeddings $\mu_{Q_t}$ of the marginal coresets to all lie within a convex hull defined by the exemplars $\{u_i\}_{i \in S}$.

By restricting the support of our coresets in this manner, we obtain data summaries that are easily comparable. Within a single dataset, we can look at the weighted exemplars that make up the coreset and use these to understand the spread of the data, as is the case with an independent MMD coreset. Indeed, since $\mathbb{Q}_t$ still meets the definition of an MMD coreset for $X_t$ (see Equation (3)), we can use it analogously to an independently generated coreset. However, since the exemplars are shared across datasets, we can directly compare the exemplars for two datasets. We no longer need to intuit similarities between disjoint sets of exemplars and their corresponding weights; instead we can directly compare the weights for each exemplar to determine their relative relevance to each dataset. We will show in Section 4.2.1 that this facilitates qualitative comparison between the marginal coresets, when compared to independently generated coresets.

We note that the dependent MMD coresets introduced in this paper are directly extensible to other integral probability measures; we could, for example, construct a dependent version of the Wasserstein coresets introduced by [22].

### 3.3. Model Understanding and Extrapolation

As we discussed in Section 2.3, MMD coresets can be used as tools to understand the performance of an algorithm on "typical" data points. Considering how an algorithm performs on such exemplars allows the practitioner to understand failure modes of the algorithm, when applied to the data. In classification tasks where labeling is expensive, or on qualitative tasks such as image modification, looking at an appropriate coreset can provide an estimate of how the algorithm will perform across the dataset.

In a similar manner, dependent coresets can be used to understand generalization behavior of an algorithm. Assume a machine learning algorithm has been trained on a given dataset $X_a$, but we wish to apply it (without modification) to a dataset $X_b$. This is frequently done in practice, since many machine learning algorithms require large training sets and high computational cost; however if the training distribution differs from the deployment distribution, the algorithm may not perform as intended. In general, we would expect the algorithm to perform well on data points in $X_b$ that have many close neighbors in $X_a$, but perform poorly on data points in $X_b$ that are not well represented in $X_a$.

Creating a dependent MMD coreset $(\mathbb{Q}_a = \sum_i w_{a,i} \delta_{u_i}, \mathbb{Q}_b = \sum_i w_{b,i} \delta_{u_i})$ for the pair $(X_a, X_b)$ allows us to identify exemplars that are highly representative of $X_a$ or $X_b$ (i.e., have high weight in the corresponding weighted coreset). Further, by comparing the weights in the two coreset measures—e.g., by calculating $f_i = w_{b,i}/w_{a,i}$ – we can identify

exemplars that are much more representative of one dataset than another. Rather than look at all points in the coreset, if we are satisfied with the performance of our model on $X_a$, we can choose to only look at points with high values of $f_i$—points that are representative of the new dataset $X_b$, but not the original dataset $X_a$. Further, if we wish to consider generalization to multiple new datasets, a shared set of exemplars reduces the amount of labeling or evaluation required.

An MMD coreset, dependent or otherwise, will only contain exemplars that are representative of the dataset(s). There are likely to be outliers that are less well represented by the coreset. Such outliers are likely to be underserved by a given algorithm—for example, yielding low accuracy or poor reconstructions.

As we saw in Section 2.4, MMD coresets can be augmented by criticisms—points in $X$ that are poorly approximated by $\mathbb{Q}$. We can equivalently construct criticisms for each dataset represented by a dependent MMD coreset. In the example above, we would select criticisms for the dataset $X_b$ by selecting points in $X_b$ that maximize

$$\hat{f}_t(x^*) = \frac{1}{n_t} \sum_{i=1}^{n_t} k(x^*, X_{t,i}) - \sum_{j \in S} w_{t,j} k(x^*, u_j).$$

In addition to evaluating our algorithm on the marginal dependent coreset for dataset $X_b$, or the subset of the coreset with high values of $f_i$, we can evaluate on the criticisms $C_b$. In conjunction, the dependent MMD coreset and its criticisms allow us to better understand how the algorithm is likely to perform on both typical, and atypical, exemplars of $X_b$.

### 3.4. A Greedy Algorithm for Finding Dependent Coresets

Given a collection $\{X_t\}_{t \in \mathcal{T}}$ of datasets, where we assume $X_t := \{x_{t,1}, \ldots, x_{t,n_t}\} \sim \mathbb{P}_t$, and a set of $n_U$ candidates $U$, our goal is to find a collection $\{\mathbb{Q}_t\}_{t \in \mathcal{T}}$ with shared support $\{u_i : i \in S \subset [n_U]\}$ such that $\widehat{\mathrm{MMD}^2}(X_t, \mathbb{Q}_t) \leq \epsilon^2$ for all $t \in \mathcal{T}$. We begin by constructing an algorithm for a related task: to minimize $\sum_{t \in \mathcal{T}} MMD(\mathbb{Q}_t, \mathbb{P}_t)$, where

$$\widehat{\mathrm{MMD}^2}(X_t, \mathbb{Q}_t) = \frac{1}{n_t^2} \sum_{i=1}^{n_t} \sum_{j=1}^{n_t} k(x_{t,i}, x_{t,j}) + \sum_{i \in S} \sum_{j \in S} w_{t,i} w_{t,j} k(u_i, u_j) - \frac{2}{n_t} \sum_{i=1}^{n_t} \sum_{j \in S} w_{t,j} k(x_i, u_j) \quad (5)$$

where $\mathbb{Q}_t = \sum_{i \in S} w_{t,i} \delta_{u_i}$. If we ignore terms in Equation (5) that do not depend on the $\mathbb{Q}_t$, we obtain the following loss:

$$\mathcal{L}(\{\mathbb{Q}_t\}_{t \in \mathcal{T}}) = \sum_{t \in \mathcal{T}} \ell_t(\mathbb{Q}_t)$$

$$\ell_t(\mathbb{Q}_t) = \frac{1}{2} \sum_{i \in S} \sum_{j \in S} w_{t,i} w_{t,j} k(u_i, u_j) - \frac{1}{n_t} \sum_{i=1}^{n_t} \sum_{j \in S} k(x_i, u_j). \quad (6)$$

We can use a greedy algorithm to minimize this loss. Let $\mathbb{Q}_t^{(m)} = \sum_{i \in S^{(m)}} w_{t,i}^{(m)} \delta_{u_i^{(m)}}$, where $S^{(m)}$ indexes the first $m$ exemplars to be added. We wish to select the exemplar $u_*$, and set of weights $\left\{ w_{t,*}^{(m+1)}, \{w_{t,i}^{(m+1)}\}_{i \in S^{(m)}} \right\}$ for each dataset $X_t$, that minimize the loss. However, searching over all possible combinations of exemplars and weights is prohibitively expensive, as it involves a non-linear optimization to learn the weights associated with each candidate. Instead, we assume that, for each $t \in \mathcal{T}$, there is some $\alpha_t^* > 0$ such that $w_{t,*}^{(m+1)} = \frac{\alpha_t^*}{\alpha_t^* + 1}$ and $w_{t,i}^{(m+1)} = \frac{w_{t,i}^{(m)}}{\alpha_t^* + 1}$ for all $i \in S^{(m)}$. In other words, we assume that the relative weights in each $\mathbb{Q}_t$ of the previously added exemplars do not change as we add more exemplars.

Fortunately, the value of $\alpha_t^*$ that minimizes $\ell_t\left(\frac{1}{1+\alpha_t^*}\mathbb{Q}_t^{(m)} + \frac{\alpha_t^*}{\alpha_t^*+1}\delta_{u_*}\right)$ can be found analytically for each candidate $u_*$ by differentiating the loss in step 1, yielding

$$\alpha_t^* = \frac{\displaystyle\sum_{i,j\in S^{(m)}} w_{t,i}w_{t,j}k(u_i,u_j) - \sum_{i\in S^{(m)}} w_{t,i}k(u_i,u_*) - \frac{1}{n_t}\sum_{i=1}^{n_t}\left(k(x_i,u_*) - \sum_{j\in S^{(m)}} w_j k(x_{t,i},u_j)\right)}{k(u_*,u_*) - \displaystyle\sum_{i\in S^{(m)}} w_{t,i}k(u_i,u_*) - \frac{1}{n_t}\sum_{i=1}^{n_t}\left(k(x_i,u_*) - \sum_{j\in S^{(m)}} w_j k(x_{t,i},u_j)\right)}. \quad (7)$$

We can, therefore, set

$$i^*, \{\alpha_t^*\} \leftarrow \underset{\substack{i\in[n_U]\setminus S^{(m)},\, t\in\mathcal{T} \\ \alpha_t\in\mathbb{R}_+}}{\arg\min} \sum \ell_t\left(\frac{1}{1+\alpha_t}\mathbb{Q}_t^{(m)} + \frac{\alpha_t}{\alpha_t+1}\delta_{u_i}\right) \quad (8)$$

and let $S^{(m+1)} = S^{(m)} \cup i^*$, $w_{t,i^*}^{(m+1)} = \frac{\alpha_t^*}{\alpha_t^*+1}$ for all $t\in\mathcal{T}$, and $w_{t,i}^{(m+1)} = \frac{w_{t,i}^{(m)}}{\alpha_t^*+1}$ for all $t\in\mathcal{T}$ and $i\in S^{(m)}$.

As written, the procedure will greedily minimize the sum of the per-dataset losses. However, the definition of an MMD coreset involves satisfying, not minimizing: we want $\widehat{\mathrm{MMD}^2}(X_t,\mathbb{Q}_t) \le \epsilon^2$ for all $t\in\mathcal{T}$. To achieve this, we modify the sum in Equation (8) so that it only includes terms for which $\widehat{\mathrm{MMD}^2}(X_t,\mathbb{Q}_t^{(m)}) > \epsilon^2$. The resulting procedure is summarized in Algorithm 1.

---

**Algorithm 1** DMMD: Selecting dependent MMD coresets

---

**Require:** Datasets $\{X_t\}_{t\in\mathcal{T}}$; candidate set $U$; kernel $k(\cdot,\cdot)$; threshold $\epsilon^2 > 0$
  $S^{(0)} \leftarrow \varnothing$; $w_t^{(0)} \leftarrow [\,]$ for all $t\in\mathcal{T}$; $D\leftarrow\mathcal{T}$; $m\leftarrow 0$
  **while** $D \ne \varnothing$ **do**
    **for all** $i\in[n_U]\setminus S^{(m)}$ **do**
      **for all** $t\in\mathcal{T}$ **do**
        Calculate $\alpha_{t,i}^*$ using Equation (7)
      **end for**
      $L_i \leftarrow 0$
      **for all** $t\in D$ **do**
        $L_i \leftarrow L_i + \ell_t\left(\frac{1}{1+\alpha_t^*}\mathbb{Q}_t^{(m)} + \frac{\alpha_t^*}{\alpha_t^*+1}\delta_{u_*}\right)$
      **end for**
    **end for**
    $i^* = \arg\min_{i\in[n_U]\setminus S^{(m)}} L_i$
    $S^{(m+1)} \leftarrow S^{(m)} \cup \{i^*\}$
    $w_{i^*}^{(m+1)} \leftarrow \frac{\alpha_{i^*}}{\alpha_{i^*}+1}$
    **for all** $t\in\mathcal{T}$ **do**
      **for all** $i\in S^{(m)}$ **do**
        $w_i^{(m+1)} \leftarrow \frac{w_i^{(m)}}{\alpha_{i^*}+1}$
      **end for**
      $\mathbb{Q}_t^{(m+1)} \leftarrow \sum_{i\in S^{(m+1)}} w_{t,i}^{(m+1)}\delta_{u_i}$
    **end for**
    $D = \{X_t : \widehat{\mathrm{MMD}^2}(X_t,\mathbb{Q}_t^{(m+1)}) > \epsilon^2\}$
    $m \leftarrow m+1$
  **end while**

---

### 3.5. Limitations

As discussed in Section 2, if we can bound the MMD between two distributions by $\epsilon$, then for any function $f$ in the unit ball $\mathcal{H}$ of the Hilbert space associated with our kernel,

the expectations of $f$ with respect to the two distributions will differ by at most $\epsilon$. However, we have no guarantee for functions that cannot be represented in that Hilbert space. If we use an MMD coreset (dependent or otherwise) to understand the output of a model, and that output cannot be well approximated by the expectation with respect to a function in $\mathcal{H}$, we cannot use performance on the coreset to bound performance on the full dataset. For this reason, we focus on the use of coresets as a qualitative, diagnostic tool for exploring model performance.

Beyond the question of whether functions of interest lie in a Hilbert space, we must also question *which* Hilbert space. Our choice of kernel will impact the nature of the resulting coresets. If we assume the popular squared exponential kernel, then different lengthscales will cause the algorithm to prioritize capturing variation at different scales. In this work, we have used median heuristics to set the lengthscale [38]; however if we were interested in capturing differences on a specific task, a better approach might be to learn the kernel. An alternative approach would be to use a different integral probability metric in place of the MMD, such as the Wasserstein distance, which has been used to construct (non-dependent) measure coresets [22].

Conversely, a limitation of MMD is that calculating $\widehat{\mathrm{MMD}^2}(X, \mathbb{Q})$ scales cubically with the size of the data. Similarly, calculating the Wasserstein distance is typically computationally expensive, as in general it requires solving a linear programming problem. This limits the scalability of our algorithm; however, since subsampling a dataset yields a valid MMD coreset with high probability [27]), our algorithm could be used on samples from larger datasets. Similarly, we could replace our initial datasets with (non-dependent) MMD coresets obtained using an existing algorithm [14–17], although this would be more expensive than random sampling. In either setting, we would need to incorporate the approximation error of the random sample or coreset, into our overall approximation error $\epsilon$.

When working with complex datasets such as images, we often work with lower-dimensional representations or embeddings [39–41]—for example, in Section 4, we will use ResNet [42] to generate embeddings for yearbook photos. However, this can make notions of "similarity" opaque, since the representations can capture properties of the image that are not immediately obvious to the viewer, or do not register as important [43]. Concerningly, recent research has suggested that image representations can encode harmful human-like biases [44].

Our algorithm greedily constructs dependent coresets. Recent work on MMD coresets has found that discrepancy-based algorithms, where the full dataset is successively divided based on some discrepancy measure, can obtain smaller $\epsilon$-coresets than greedy methods or random sampling [15–17]. Unfortunately, it is not clear how to extend such a partitioning algorithm to the dependent setting; however, these results suggest that it is worth exploring alternative constructions for dependent coresets.

## 4. Experimental Evaluation

In Section 3.2, we introduced dependent MMD coresets, a summarization technique designed to allow easy comparison between related datasets, and proposed a greedy algorithm to construct dependent MMD coresets in Section 3.4. We also described, in Section 3.3, how dependent MMD coresets can be used to understand performance of models and algorithms, particularly in the context of generalization to new datasets.

In this section, we will empirically evaluate the performance of our algorithm in Section 4.1. Previous greedy algorithms for weighted MMD coresets (without dependence) proceed by first selecting a new exemplar, and then updating weights once the exemplar has been added to the coreset. While such an approach could be adapted to the dependent setting, we show that our algorithm (Algorithm 1), which pre-selects weights based on a single calculation, achieves comparable coresets with lower computational cost.

After evaluating the algorithm used to select the coresets, we will go on to explore the coresets themselves, in Section 4.2. We begin by showing how, when comparing two datasets, the shared support offered by dependent MMD coresets allows for easier

comparison than two standard MMD coresets. We then go on to show, in an example
comparing 12 related datasets, that dependent MMD coresets can allow us to capture
trends and similarities in an interpretable manner.

In Section 4.3, we turn our attention to coresets for model understanding. Here, we
simulate a scenario where we wish to deploy algorithms trained on one dataset, to a slightly
different datasets. By looking at performance on exemplars that are highly weighted in the
second dataset, but not the first, we can obtain qualitative insights on the generalization
properties of the algorithms. Adding evaluation on criticisms of the dependent MMD
coreset leads to a deeper understanding of the model behavior.

### 4.1. Evaluation of Dependent MMD Coreset Algorithm

In Section 3.4, we proposed a greedy algorithm for selecting dependent MMD coresets
(Algorithm 1, which we will denote DMMD). This algorithm selected weights (one for
each dataset) for each candidate data point, and then greedily selected a data point and
its associated weights. Since dependent coresets are introduced in this work, there is no
direct comparison algorithm; however, a natural alternative would have been to adapt
PROTODASH, an existing greedy algorithm for weighted MMD coresets, to the dependent
setting. Such an approach differs from Algorithm 1 in that weights are optimized *after* a
candidate has been selected.

Below, we review the PROTODASH algorithm, and introduce two alternative greedy
algorithms for dependent MMD coresets: a dependent version of PROTODASH, that selects
unweighted candidates then optimizes weights; and a hybrid algorithm that pre-selects
weights for candidate points, but further optimizes them after an exemplar has been added
to the coreset. We quantitatively compare these variants with Algorithm 1, showing that
pre-selecting weights provides comparable coresets to methods that optimize weights, at a
much lower computational cost.

The PROTODASH algorithm [13] for weighted MMD coresets greedily selects exemplars
that minimize the gradient of the loss in Equation (6) (for a single dataset). Having selected
an exemplar to add to the coreset, PROTODASH then uses an optimization procedure to find
the weights that minimize $\widehat{\text{MMD}^2}(X, \mathbb{Q}^{(m+1)})$. We modify this algorithm for the dependent
MMD setting by summing the gradients across all datasets for which the $\epsilon^2$ threshold is
not yet satisficed, leading to the dependent PROTODASH algorithm shown in Algorithm 2.

Unlike the dependent version of `protodash` in Algorithm 2, our algorithm assigns
weights *before* selection, which should encourage adding points that would help some
of the marginal coresets, but not others. However, there is no post-exemplar-addition
optimization of the weights. Inspired by the post-addition optimization in PROTODASH,
we also compare our algorithm with a variant of Algorithm 1 that optimizes the weights
after each step—allowing the relative weights of the exemplars to change between each
iteration. We will refer to this variant of DMMD with post-exemplar-addition optimization
as DMMD-OPT.

---

**Algorithm 2** A dependent `protodash` algorithm

---

**Require:** Datasets $\{X_t\}_{t\in\mathcal{T}}$, candidate set $U$, kernel $k(\cdot,\cdot)$, threshold $\epsilon^2 > 0$

$\quad S^{(0)} \leftarrow \varnothing, w_t^{(0)} \leftarrow []$ for all $t \in \mathcal{T}, D \leftarrow \mathcal{T}, m \leftarrow 0$

$\quad$**for all** $i \in [n_U]$ **do**

$\quad\quad g_i = \sum_{t\in\mathcal{T}} \frac{1}{n_t} \sum_{j=1}^{n_t} k(x_j, u_i)$

$\quad$**end for**

$\quad$**while** $D \neq \varnothing$ **do**

$\quad\quad i^* = \arg\min_{i\in[n_U]\setminus S^{(m)}} g_i$

$\quad\quad S^{(m+1)} \leftarrow S^{(m)} \cup \{i^*\}$

$\quad\quad$**for all** $t \in \mathcal{T}$ **do**

$\quad\quad\quad \{w_{t,i}^{(m+1)}\}_{i\in S^{(m+1)}} \leftarrow \arg\max_{\{w_{t,i}\}_{i\in S^{(m+1)}}} \ell_t(\sum_{i\in S^{(m+1)}} w_{t,i}\delta_{u_i})$

$\quad\quad$**end for**

$\quad\quad \mathbb{Q}_t^{(m+1)} \leftarrow \sum_{i\in S^{(m+1)}} w_{t,i}^{(m+1)} \delta_{u_i}$

$\quad\quad$**for all** $i \in [n_U] \setminus S^{(m+1)}$ **do**

$\quad\quad\quad g_i = \sum_{t\in D} \nabla\ell_t(\mathbb{Q}^{(m+1)})$

$\quad\quad$**end for**

$\quad\quad D = \{X_t : \widehat{\text{MMD}^2}(X_t, \mathbb{Q}_t^{(m+1)}) > \epsilon^2\}$

$\quad\quad m \leftarrow m + 1$

$\quad$**end while**

---

We evaluate all three methods using a dataset of photographs of 15,367 female-identified students, taken from yearbooks between 1905 and 2013 [45]. We show a random subset of these images in Figure 3. We generated 512-dimensional embeddings of the photos using the torchvision pre-trained implementation of ResNet [42,46]. We then partitioned the collection into 12 datasets, each containing photos from a single decade.

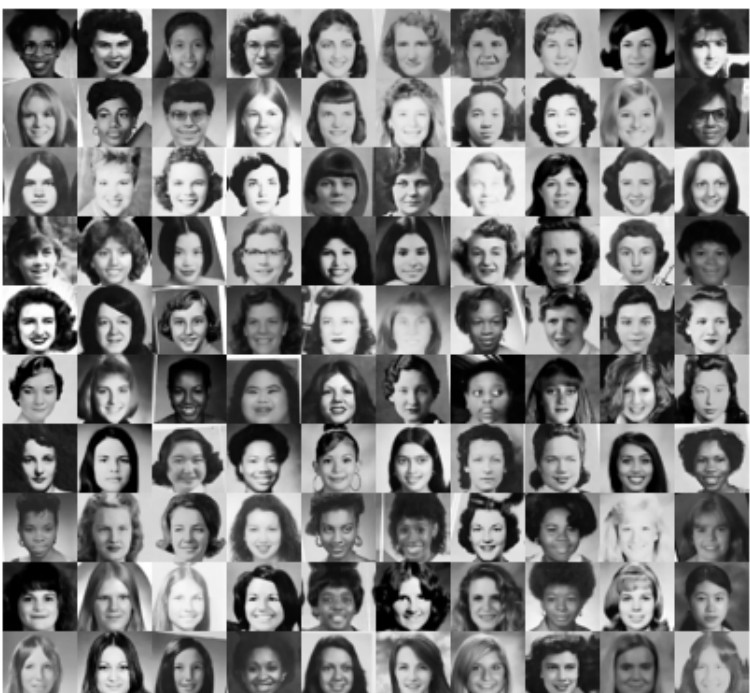

**Figure 3.** A random subset of 100 images taken from the yearbook dataset.

In order to capture lengthscales appropriate for the variation in each decade, we use an additive kernel, setting

$$K = \frac{K_{all} + \sum_{t\in\mathcal{T}} K_t}{|\mathcal{T}| + 1},$$

where $K_{all}$ is a squared exponential kernel with bandwidth given by the overall median pairwise distances; $\mathcal{T}$ is the set of decades that index the datasets; $K_t$ is a squared exponential kernel with bandwidth given by the median pairwise distance between images in dataset $t$.

We begin by considering how good a dependent MMD coreset each algorithm is able to construct, for a given number of exemplars $m = |S|$. To do so, we ran all algorithms without specifying a threshold $\epsilon^2$, recording $\widehat{\mathrm{MMD}^2}(X_t, \mathbb{Q}_t^{(m)})$ for each value of $m$. All algorithms were run for one hour on a 2019 Macbook Pro (2.6 GHz 6-Core Intel Core i7, 32 GB 2667 MHz DDR4), excluding time taken to generate and store the kernel entries, which only occurs one time. As much code as possible was re-used between the three algorithms. Where required, optimization of weights was carried out using a BFGS optimizer. Code is available at https://github.com/sinead/dmmd (accesson 22 September 2021).

Figure 4a shows the per-dataset estimates $\widehat{\mathrm{MMD}^2}(X_t, \mathbb{Q}_t^{(m)})$, and Figure 4b shows the average performance across all 12 datasets. We see that the three algorithms perform comparably in terms of coreset quality. DMMD-OPT seems to perform slightly better than DMMD, as might be expected due to the additional optimization step. PROTODASH, by comparison, seems to perform slightly worse, which we hypothesise is because it has no mechanism by which weights can be incorporated at selection time. However, in both cases, the difference is slight.

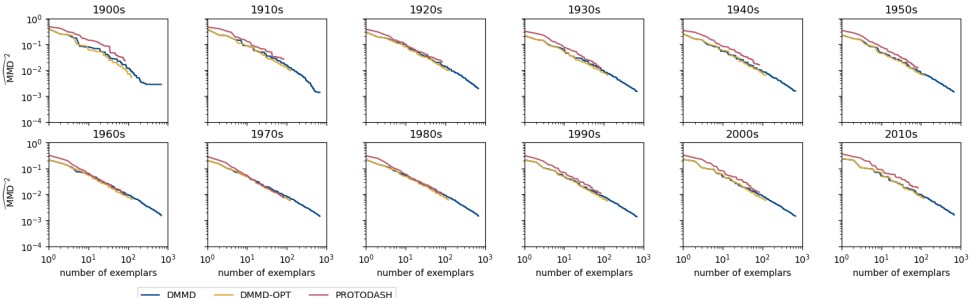

(**a**) $\widehat{\mathrm{MMD}^2}\left(X_t, \mathbb{Q}_t^{(m)}\right)$ for increasing numbers of exemplars $m$. Each plot corresponds to a dataset containing yearbook photos from a single decade.

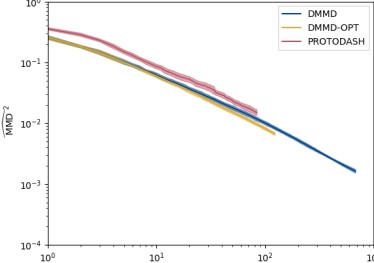

(**b**) Mean $\pm$ one standard error of $\widehat{\mathrm{MMD}^2}\left(X_t, \mathbb{Q}_t^{(m)}\right)$ for increasing numbers of exemplars $m$.

**Figure 4.** Evaluating how coreset quality varies with number of exemplars, for dependent MMD coresets generated using three algorithms, on 12 yearbook datasets.

DMMD is however *much* faster at generating coresets, since it does not optimize the full set of weights at each iteration. This can be seen in Figure 5, which shows the time taken to generate coresets of a given size. The cost of the optimization-based algorithms grows rapidly with coreset size ($m$); the rate of growth of the DMMD coresets is much smaller.

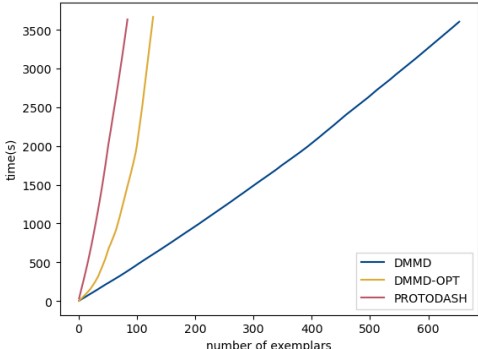

**Figure 5.** Time (in seconds) taken to construct MMD dependent coresets of a given size, for three algorithms, on the 12 yearbook datasets. Algorithms ran for a maximum of one hour.

In practice, rather than endlessly minimizing $\sum_{t \in \mathcal{T}} \widehat{\text{MMD}^2}(X_t, \mathbb{Q}_t)$, we will aim to find $\mathbb{Q}_t$ such that $\widehat{\text{MMD}^2}(X_t, \mathbb{Q}_t) < \epsilon^2$ for all $t \in \mathcal{T}$. In Figure 6, we show the coreset sizes required to obtain an $\epsilon$-MMD dependent coreset on the twelve decade-specific yearbook datasets, for each algorithm. Again, a maximum runtime of one hour was specified. When all three algorithms were able to finish, the coreset sizes are comparable (with DMMD-OPT finding slightly smaller coresets than DMMD, and PROTODASH finding slightly larger coresets). However the optimization-based methods are hampered by their slow runtime.

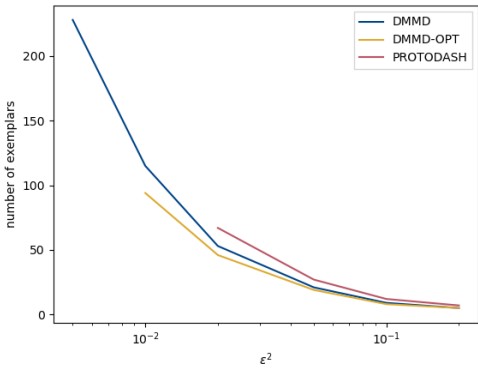

**Figure 6.** Coreset size required to obtain an $\epsilon$-MMD dependent coreset on the 12 yearbook datasets, for three algorithms. Algorithms ran for a maximum of one hour; PROTODASH failed to complete coresets for $\epsilon^2 = 0.01$ and $\epsilon^2 = 0.005$. DMMD-OPT failed to complete a coreset for $\epsilon^2 = 0.005$.

Based on these analyses, it appears there is some advantage to additional optimization of the weights. However, in most cases, we do not feel the additional computational cost merits the improved performance.

### 4.2. Interpretable Data Summarizations

Summarizations of datasets can allow us to quickly understand properties of their distributions, and allow us to convey such properties to others, for example in a document explaining the data and its providence [47,48]. In high-dimensional, highly structured datasets such as collections of images, traditional summary statistics such as the mean of a dataset are particularly uninterpretable, as they convey little of the shape of the underlying distribution. A better approach is to show the viewer a collection of images that are representative of the dataset. MMD coresets allow us to obtain such a representative set, making them a better choice than displaying a random subset.

As we discussed in Section 3.1, if we wish to summarize a collection of related datasets, independently generated MMD coresets can help us understand each dataset individually, but it may prove challenging to compare datasets. This challenge becomes greater in high dimensional settings such as image data, where we cannot easily intuit a distance

between exemplars. To showcase this phenomenon, and demonstrate how dependent MMD coresets can help, we return to the yearbook photos introduced in Section 4.1. For all experiments in this section, we use the additive kernel described in Section 4.1.

4.2.1. A Shared Support Allows for Easier Comparison of Datasets

In Section 3.2, we argued that the shared support provided by dependent MMD coresets facilitates comparison of datasets, since we only need to consider differences in weights. To demonstrate this, we constructed four datasets, each a subset of the entire yearbook dataset containing 250 photos. The first two datasets contained only faces from the 1990s; the second two, only faces from the 2000s. The datasets were generated by sampling without replacement from the associated decades, to ensure no photo appeared more than once across the four datasets. Our goal is to provide a visual way to compare these four datasets.

We begin by independently generating (non-dependent) MMD coresets for the four datasets, using Algorithm 1 independently on each dataset, with a threshold of $\epsilon^2 = 0.01$. The set of candidate images, $U$, was the entire dataset of 15,367 images. The resulting coresets are shown in Figure 7; the areas of the bubbles correspond to the weights associated with each exemplar (The top row of Figure 2 duplicates Figure 7).

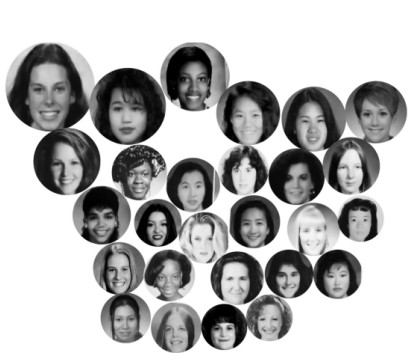

(**a**) 0.1-MMD coreset for a set of 250 randomly selected yearbook photos from the 1990s.

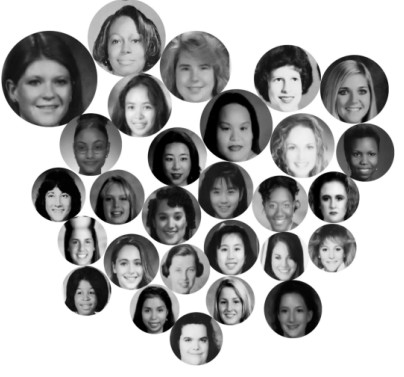

(**b**) 0.1-MMD coreset for a second set of 250 randomly selected yearbook photos from the 1990s.

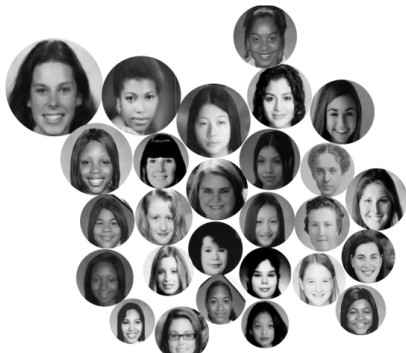

(**c**) 0.1-MMD coreset for a set of 250 randomly selected yearbook photos from the 2000s.

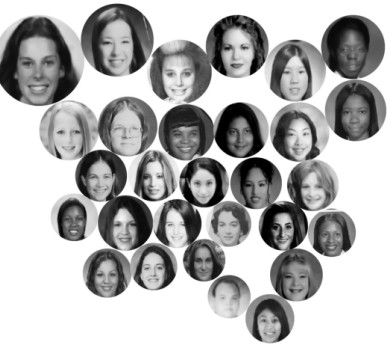

(**d**) 0.1-MMD coreset for a second set of 250 randomly selected yearbook photos from the 2000s.

**Figure 7.** Independently learned, weighted 0.1-MMD coresets based on 250 random samples from a given decade. Area of each bubble is proportional to the weight of the corresponding exemplar in the coreset.

We can see that, considered individually, each coreset appears to be doing a good job of capturing the variation in students for each dataset. However, if we compare the four

coresets, it is not easy to tell that Figure 7a,b represent the same underlying distribution, and Figure 7c,d represent a second underlying distribution—or to interpret the difference between the two distributions. We see that the highest weighted exemplar for the two 2000s datasets is the same(top left of Figure 7c,d), but only one other image is shared between the two coresets. Meanwhile, the first coreset for the 1990s shares the same highest-weighted image with the two 2000s datasets—but this coreset does not appear in the first 1990s coreset, and the two 1990s coresets have no overlap. Overall, it is hard to compare between the marginal coresets.

By contrast, the shared support offered by dependent coresets means we can directly compare the distributions using their coresets. In Figure 8, we show a dependent MMD coreset ($\epsilon^2 = 0.01$) for the same collection of datasets. The shared support allows us to see that, while the two decades are fairly similar, there is clearly a stronger similarity between the pairs of datasets from the same year (i.e., similarly sized photos), than between pairs from different years. We can also identify images that exemplify the difference between the two decades, by looking at the difference in weights. We see that many of the faces towards the top of the bubble plot have high weights in the 2000s, but low weights in the 1990s. Examining these exemplars suggests that straight hair became more prevalent in the 2000s. Conversely, many of the faces towards the bottom of the bubble plot have high weights in the 1990s, but low weights in the 2000s. These photos tend to have wavy/fluffy hair and bangs. In conjunction, these plots suggest a tendency in the 2000s away from bangs and towards straight hair, something the authors remember from their formative years. However, there is still a significant overlap between the two decades: many of the exemplars have similar weights in the 1990s and the 2000s.

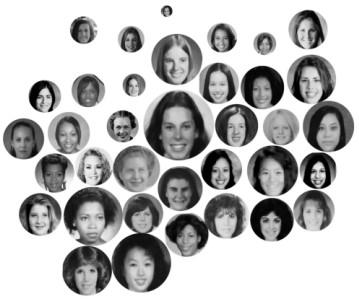

(**a**) Marginal dependent 0.1-MMD coreset for a set of 250 randomly selected yearbook photos from the 1990s.

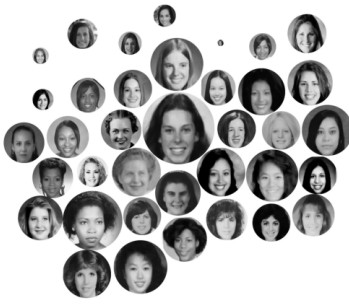

(**b**) Marginal dependent 0.1-MMD coreset for a second set of 250 randomly selected yearbook photos from the 1990s.

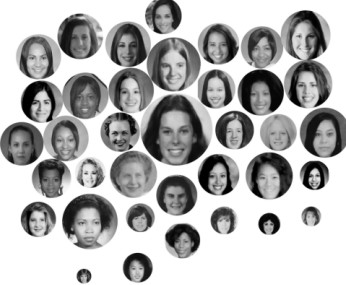

(**c**) Marginal dependent 0.1-MMD coreset for a set of 250 randomly selected yearbook photos from the 2000s.

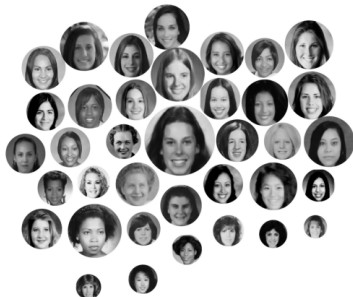

(**d**) Marginal dependent 0.1-MMD coreset for a second set of 250 randomly selected yearbook photos from the 2000s.

**Figure 8.** Dependent 0.1-MMD coreset for a collection of 4 datasets, each including 250 random samples from a given decade. Area of each bubble is proportional to the weight of the corresponding exemplar in the marginal coreset. Positioning is constant across all four examples.

We can also see this in Figure 9, a bar chart shows the average weights associated with each exemplar in each decades (i.e., the blue bar above a given image is the average weight for that exemplar across the two 1990s datasets, and the red bar is the average weight across the two 2000s datasets). We see that most of the exemplars have similar weights in both scenarios, but that we have a number of straight-haired exemplars disproportionally representing the 2000s, and a number of exemplars with bangs and/or wavy hair disproportionately representing the 1990s. These insights would have been hard to intuit from the standard MMD coresets, where it is hard to identify what variation is due to true underlying differences in the dataset, and what is due to sampling error.

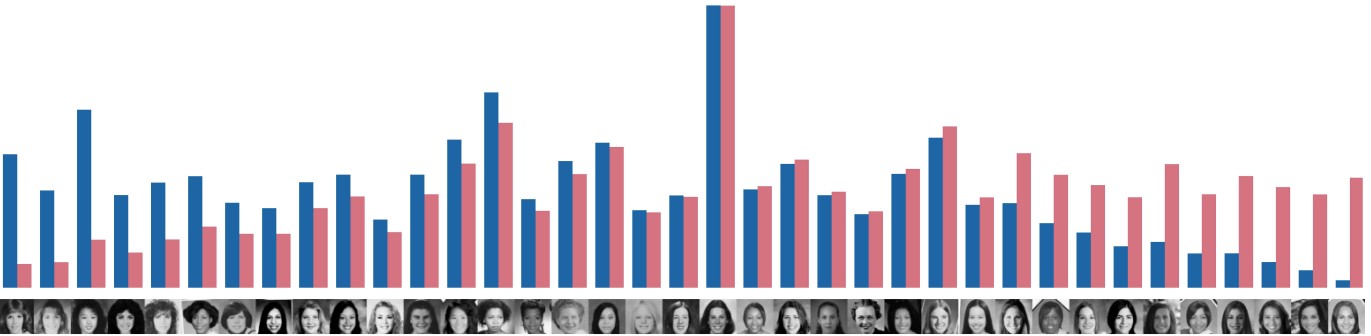

**Figure 9.** Summary of a dependent 0.1-MMD coreset for four datasets of yearbook faces from the 1990s and 2000s. Exemplars are shown along the $x$ axis. The average weight for each exemplar in the coresets associated with the 1990s is shown in blue ▬; the average weight for the 2000s is shown in red ▬.

### 4.2.2. Dependent Coresets Allow Us to Visualize Data Drift in Collections of Images

Next, we show how dependent MMD coresets can be used to understand and visualize variation between collections of multiple datasets. As in Section 4.1, we partition the 15,367 yearbook images into twelve datasets based on their decade, with the goal of understanding how the distribution over yearbook photos changes over time. Table 3 shows the number of photos in each resulting dataset.

**Table 3.** Number of yearbook photos for each decade.

| 1900s | 1910s | 1920s | 1930s | 1940s | 1950s | 1960s | 1970s | 1980s | 1990s | 2000s | 2010s |
|---|---|---|---|---|---|---|---|---|---|---|---|
| 35 | 98 | 308 | 1682 | 2650 | 2093 | 2319 | 2806 | 2826 | 2621 | 2208 | 602 |

Figure 10 shows the exemplars in the resulting dependent MMD coreset, with a threshold of $\epsilon^2 = 0.01$. The corresponding plots show how the weights vary with time. The exemplars are ordered based on their average weight across the 12 datasets. In each case, a red, vertical line indicates the year of the yearbook from which the exemplar was taken. We are able to see how styles change over time, moving away from the formal styles of the early 20th century, through waved hairstyles popular in the midcentury, towards longer, straighter hairstyles in later decades. In general, the relevance of an exemplar peaks around the time it was taken (although, this information is not used to select exemplars). However, some styles remain relevant over longer time periods (see many exemplars in the first column). Most of the early exemplars are highly peaked on the 00s or 10s; this is not surprising since these pre-WW1 photos tend to have very distinctive photography characteristics and hair styles. Note that we do not include a comparison to standard, independent MMD coresets, as it would not be possible to produce an analogous set of plots—the exemplars in each decade's coreset would, in general, not overlap.

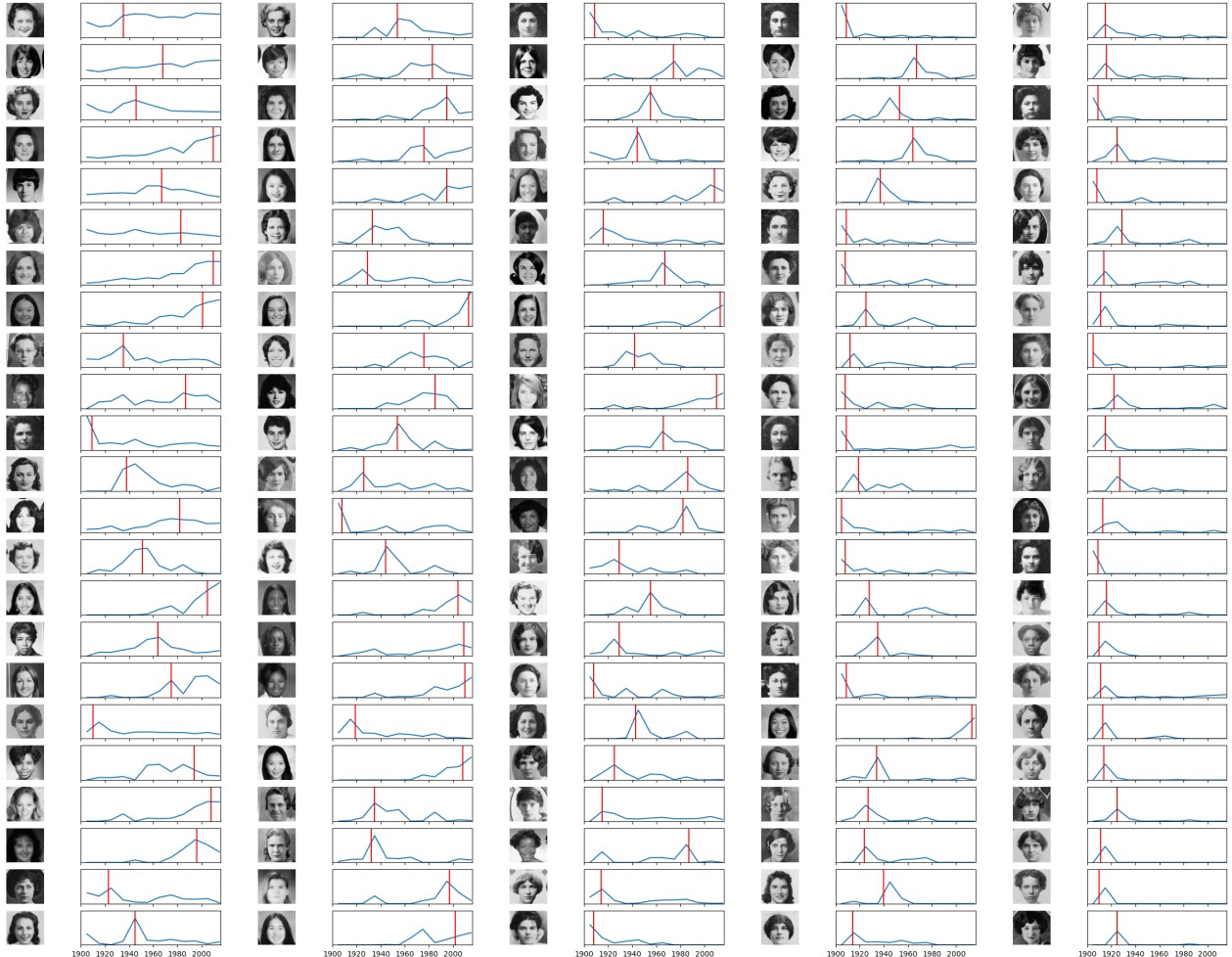

**Figure 10.** Visualization of an 0.1-MMD dependent coreset for 12 datasets, each containing yearbook photos from a given decades. Photos show the exemplars $\{u_i : i \in S\}$, ordered by their average weight across the 12 marginal coresets. To the left of each photo is a plot of the corresponding weight over time; a red vertical line marks the year the photo was taken.

Figure 10 appears to show that the marginal coresets have high weights on exemplars from the corresponding decade. To look at this in more detail, we consider the distributions over the dates of the exemplars associated with each decade. Figure 11 shows the weighted mean and standard deviation of the years associated with the exemplars, with weights given by the coreset weights. We see that the mean weighted year of the exemplars increases with the decade. However, we notice that it is pulled towards the 1940s and 1950s in each case: this is because we must represent all datasets using a weighted combination of points taken from the convex hull of all datapoints.

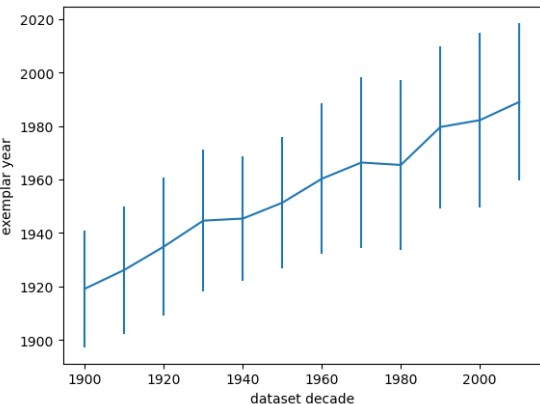

**Figure 11.** Distribution over the year associated with the marginal coresets for each decade. Plot shows weighted mean ± weighted standard deviation.

*4.3. Dependent Coresets Allow Us to Understand Model Generalization*

To see how dependent coresets can be used to understand how a model trained on one dataset will generalize to others, we simulate a scenario where we wish to deploy a machine learning model on a given dataset, but where the model was trained on a different dataset. In this scenario, we are interested in learning whether the model generalizes well to the new dataset.

We generate two datasets—one to represent the training data, and one to represent the data used in deployment—by partitioning a collection of image digits. We started with the USPS handwritten digits dataset [49], which comprises a train set of 2791 handwritten digits, and a test set of 2001 handwritten digits. We split the train set into two datasets, $X_a$ and $X_b$, where $X_a$ is skewed towards the earlier digits and $X_b$ towards the later digits. Figure 12 shows the resulting label counts for each dataset: we see there is a clear distributional imbalance. Note that, in general, we will not have such a concise summary of the difference between two datasets; however using image digits as our example allows us to get an idea of the "ground truth" difference between the two datasets.

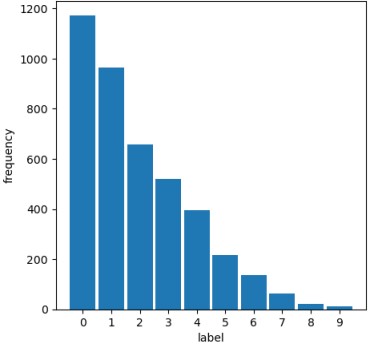

(**a**) Label frequencies for dataset $X_a$

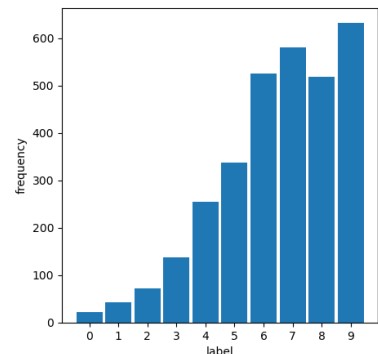

(**b**) Label frequencies for dataset $X_b$

**Figure 12.** Frequency with which each digit occurs in two datasets of handwritten digits.

We selected three classification algorithms to assess generalization performance. We chose classification algorithms because it is easy for us to obtain "ground truth" generalization performance by applying these algorithms to our second dataset $X_b$, allowing us to compare our insights with the true generalization performance. In general, we may not be able to easily estimate generalization performance in this manner: we may have unlabeled data, or our task may not be easily qualitatively evaluated (e.g., evaluating quality of auto-generated captions); we expect our approach to have greatest utility in such scenarios.

We trained three classifiers—a decision tree with maximum depth of 8, a random forest with 100 trees, and a multilayer perceptron (MLP) with a single hidden layer with

100 units—on $X_a$ and the corresponding labels. In each case, we used the implementation in scikit-learn [50], with parameters chosen to have similar train set accuracy on $X_a$. These three models were chosen to have varying generalization accuracy. Table 4 shows the associated classification accuracies on the datasets $X_a$ and $X_b$, which we will use as a quantitative representation of the algorithms' generalization performance on $X_b$. We also show confusion matrices in Figures 13 and 14. We see that all three algorithms perform comparably on $X_a$, the dataset on which they were trained. However, when applied to $X_b$, we see in Figure 14 that the decision tree struggles in classifying 8s and 9s, and that the random forest struggles with 9s.

**Table 4.** Accuracies of three classification algorithms, on datasets $X_a$ and $X_b$. All algorithms were trained on $X_a$.

| Model | Accuracy on $X_a$ | Accuracy on $X_b$ |
|---|---|---|
| MLP | 0.9998 | 0.8531 |
| Random Forest | 1.0 | 0.7129 |
| Decision Tree | 0.9585 | 0.5880 |

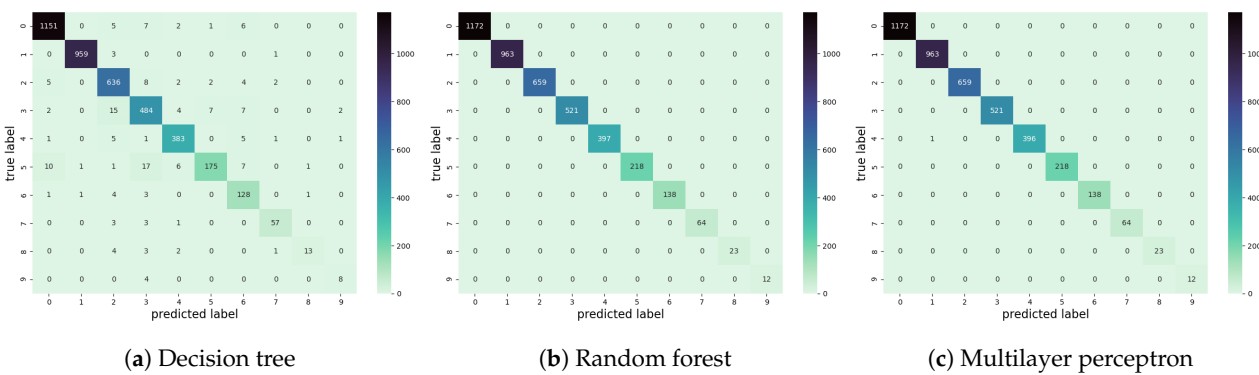

(**a**) Decision tree   (**b**) Random forest   (**c**) Multilayer perceptron

**Figure 13.** Confusion matrices on $X_a$, for three classification algorithms trained on $X_a$.

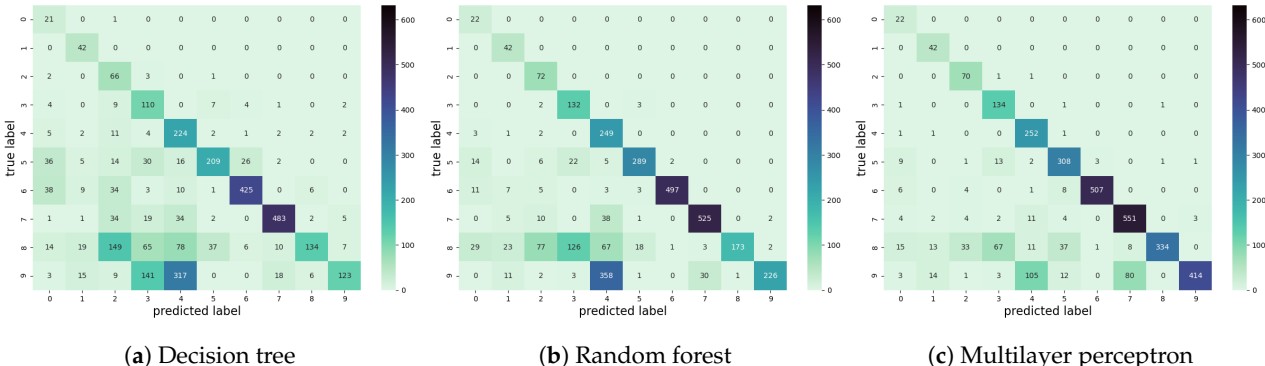

(**a**) Decision tree   (**b**) Random forest   (**c**) Multilayer perceptron

**Figure 14.** Confusion matrices on $X_b$, for three classification algorithms trained on $X_a$.

We begin our analysis by generating a dependent MMD coreset for the two datasets, with $\epsilon^2 = 0.005$. To ensure the exemplars in our coreset have not been seen in training, we let our set of candidate points $U$ be the union of $X_b$ and the USPS test set. As with the yearbook data, we use an additive squared exponential kernel, with bandwidths of the composite kernels being the median within-class pairwise distances, and the overall median pairwise distance. Distances were calculated using the raw pixel values. Figure 15 shows the resulting dependent MMD coreset, with the bars showing the weights $w_{a,i}, w_{b,i}$ associated with the two datasets, and the images below the $x$ axis showing the corresponding images $u_i$. In Figure 16, the $u_i$ and $w_i$ have been grouped by number, so that if $y(u)$ is the label of image $u$, the $j$th bar for $\mathbb{Q}_a$ has weight $\sum_{i \in S:y(u_i)=j} w_{a,j}$.

We can see that the coreset has selected points that cover the spread of the overall dataset. However, looking at Figure 16, we see that the weights assigned to these exemplars in $\mathbb{Q}_a$ and $\mathbb{Q}_b$ mirror the relative frequencies of each digit in the corresponding datasets $X_a$ and $X_b$ (Figure 12).

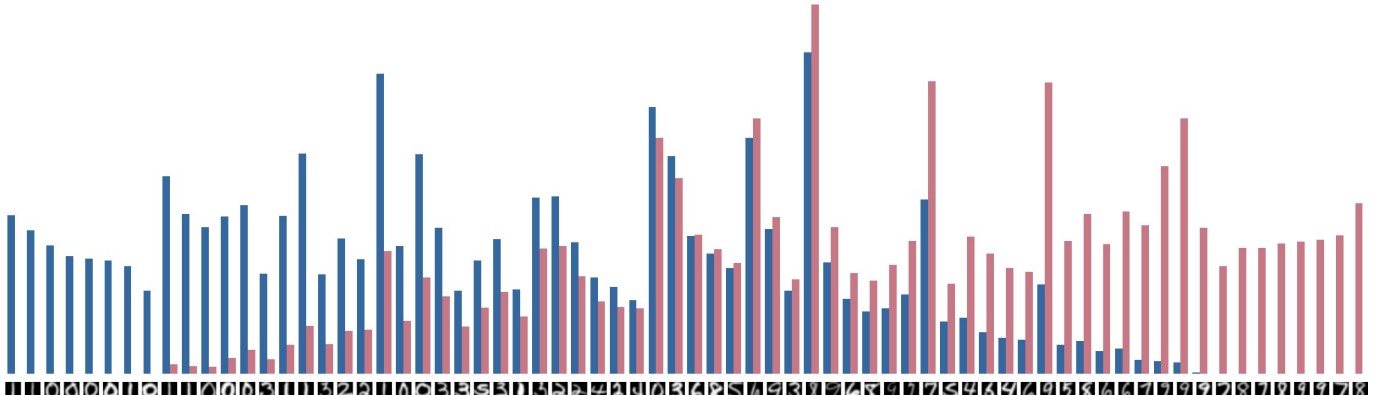

**Figure 15.** Dependent MMD coreset for two datasets of handwritten digits. Bars show the weight in each marginal coreset, with $X_a$ shown in blue ▉ and $X_b$ shown in red ▉; images along axis show corresponding exemplars.

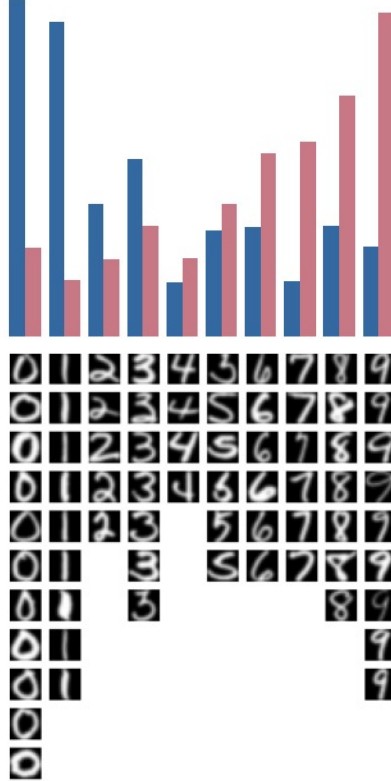

**Figure 16.** Summary of a dependent MMD coreset for two datasets of handwritten digits. The weights and exemplars from Figure 15 have been combined based on their label. Weights for $X_a$ are shown in blue ▉ and weights for $X_b$ are shown in red ▉.

We then considered all points $u_i$ in our dependent coreset $(\mathbb{Q}_a, \mathbb{Q}_b)$ where $f_i = w_{b,i}/w_{a,i} > 2$—i.e., points that are much more representative of $X_b$ than $X_a$. We then looked at the class probabilities of the three algorithms, on each of these points, as shown in Figure 17. We see that the decision tree mis-classifies nine of the 21 exemplars, and is frequently highly confident in its misclassification. The random forest misclassifies three examples, and the MLP misclassifies two. We see this agrees with the ordering

provided by empirically evaluating generalization in Table 4—the MLP generalizes best, and the decision tree worst. As suggested by our confusion matrices in Figure 14, we see that all algorithms generalize worst to the numbers 8 and 9—this is to be expected, since these digits are most under-represented in $X_a$. The decision tree in particular appears to fail on these digits, mirroring the quanitative results in the confusion matrix.

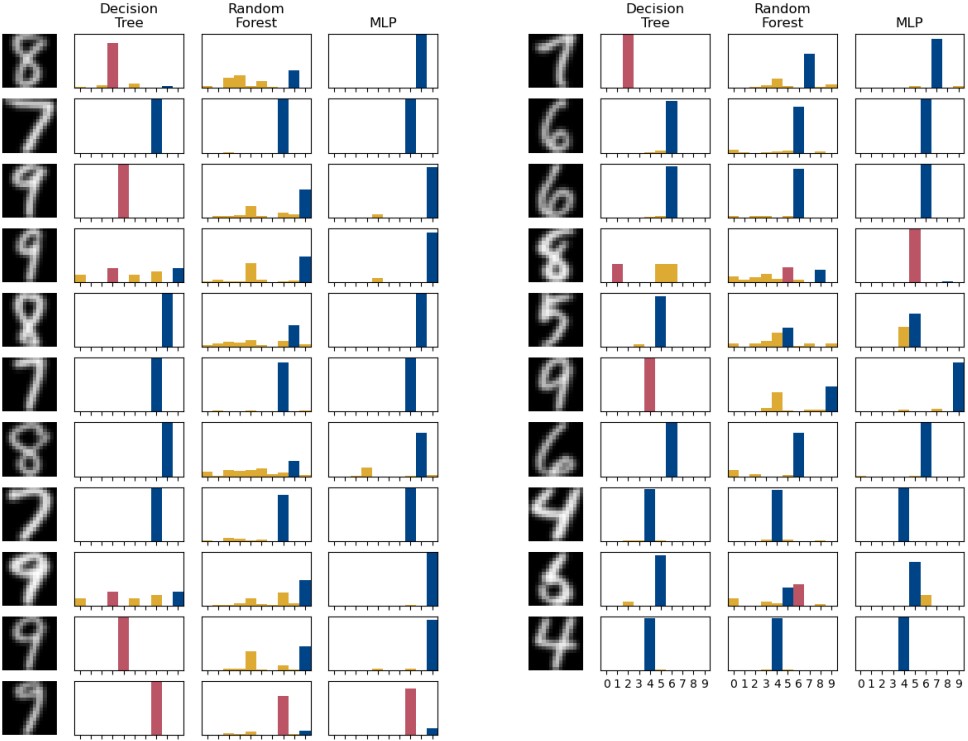

**Figure 17.** Exemplars over-represented in $\mathbb{Q}_b$, with class probabilities under three algorithms trained on $X_a$. The true class is shown in blue ▬; where the highest probability class differs from the true class, the highest probability class is shown in red ▬.

For comparison, in Figure 18 we show the points where $f_i < 0.5$—i.e., points that are much more representative of $X_a$ than $X_b$. Note that, since our candidate set did not include any members of $X_a$, none of these points were in our training set. Despite this, the accuracy is high, and fairly consistent between the three classifiers (the decision tree misclassifies two exemplars; the other two algorithms make no errors).

The dependent coreset only provides information about performance on "representative" members of $X_b$. Since classifiers will tend to underperform on outliers, looking only at the dependent MMD coreset does not give us a full picture of the expected performance. We can augment our dependent MMD coreset with criticisms—points that are poorly described by the dependent coreset. Figure 19 shows the performance of the three algorithms on a size-20 set of criticisms for $X_b$. Note that, overall, accuracy is lower than for the coreset—unsurprising, since these are outliers. However, as before, we see that the decision tree performs worst on these criticisms (nine mis-classifications), with the other two algorithms performing slightly better (six mis-classifications for the random forest, and seven for the MLP).

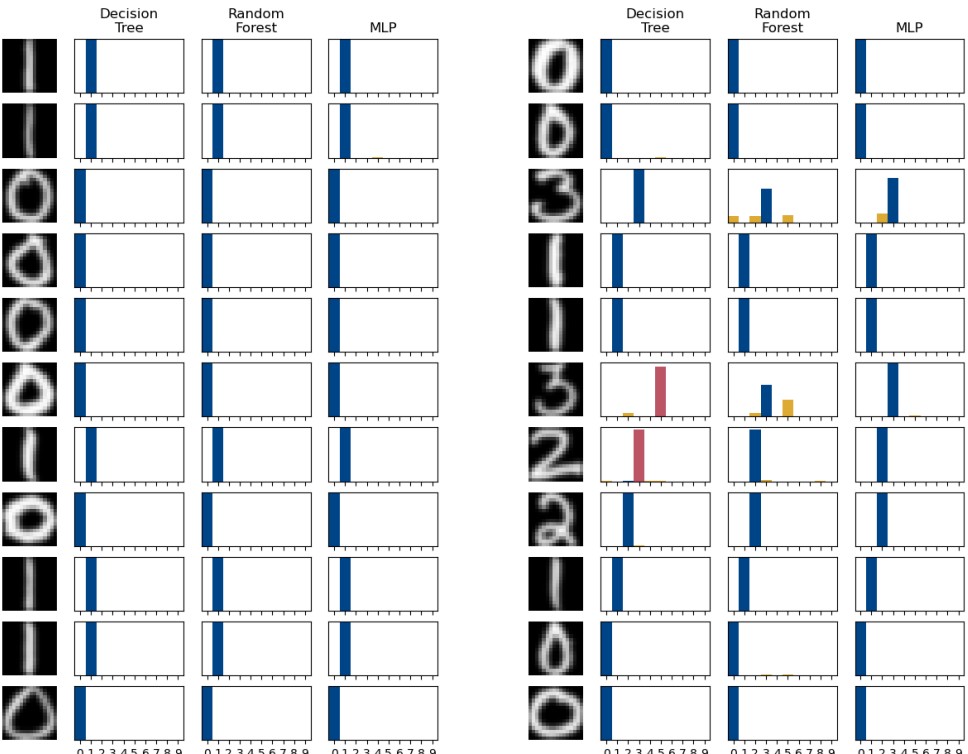

**Figure 18.** Exemplars over-represented in $\mathbb{Q}_a$, with class probabilities under three algorithms trained on $X_a$. The true class is shown in blue ██████; where the highest probability class differs from the true class, the highest probability class is shown in red ██████.

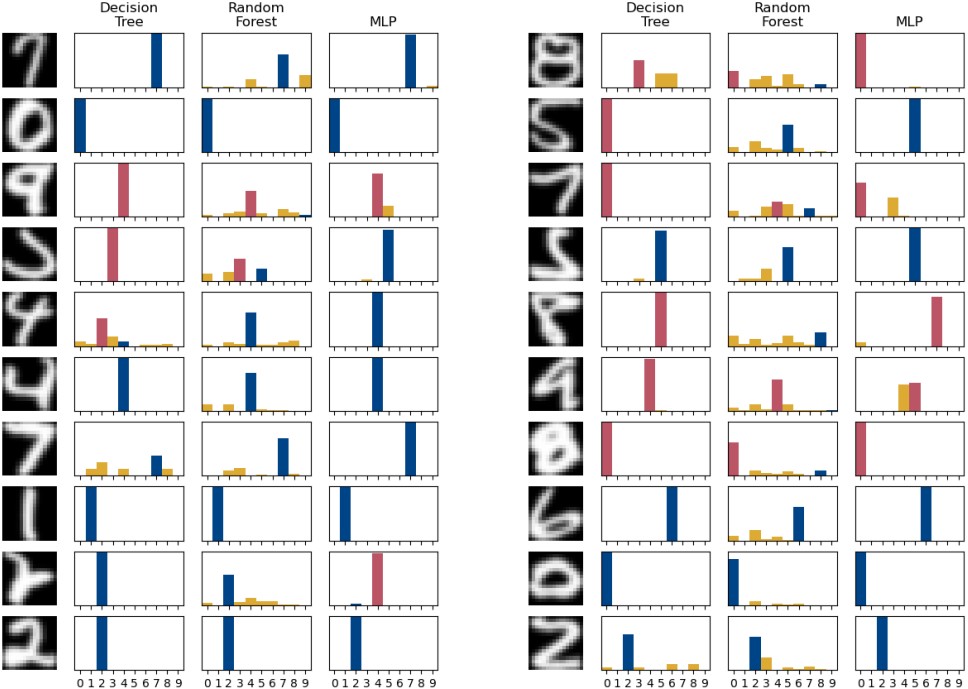

**Figure 19.** Criticisms of $\mathbb{Q}_b$ from the dataset $X_b$, with class probabilities under three algorithms trained on $X_a$. The true class is shown in blue ██████; where the highest probability class differs from the true class, the highest probability class is shown in red ██████.

Note that, since accuracy does not correspond to a function in a RKHS, we cannot expect to use the coreset to bound the expected accuracy of an algorithm on the full

dataset. Indeed, while the coresets and critics correctly suggest that the decision tree generalizes poorly, they do not give conclusive evidence on the relative generalization abilities of the other two algorithms. However, they do highlight what sort of data points are likely to be poorly modeled under each algorithm. By providing a qualitative assessment of performance modalities and failure modes on either typical points for a dataset, or points that are disproportionately representative of a dataset (vs the original training set) dependent MMD coresets allows users to identify potential generalization concerns for further exploration.

*4.4. Discussion*

MMD coresets have already proven to be a useful tool for summarizing datasets and understanding models. However, as we have shown in Section 4.2, their interpretability wanes when used to compare related datasets. Dependent MMD coresets provide a tool to jointly model multiple datasets using a shared set of exemplars. This shared set of exemplars makes it easy to compare two datasets, providing an interpretable summary not just of each dataset in isolation, but also of the difference between datasets. As such, we believe they will prove useful in understanding related datasets, and summarizing such collections of datasets.

In addition to facilitating understanding of data, we have also shown that dependent MMD coresets can be used to better understand model performance. By considering the weights associated with two different datasets, we can identify areas of domain mis-match. By exploring performance of algorithms on such points, we can glean insights about the ability of a model to generalize to new datasets.

In principle, dependent MMD coresets can be applied to any number of datasets. However, as we discuss in Section 3.5, the computational cost of our algorithm will scale cubically in the number of datapoints in the union of the datasets. This cost can be reduced by representing each dataset with an independent coreset, either obtained by subsampling the data or by applying a coreset selection algorithm such as [14–17]; however, the approximation error of this coreset would need to be incorporated into the overall approximation error $\epsilon$.

An alternative approach might be to develop streaming algorithms for constructing dependent MMD coresets. In the non-dependent setting, streaming algorithms such as [16] allow us to construct a coreset in an online manner, at a lower computational cost than batch algorithms. Such an approach would be particularly appealing in the case of time-stamped data, since it would allow us to update our dependent MMD coreset to include a new dataset.

Dependent MMD coresets are just one example of a dependent coreset that could be constructed using this framework. Future directions include exploring dependent analogues of other measure coresets [22].

**Author Contributions:** Conceptualization, S.A.W. and J.H.; methodology, software, and experiments, S.A.W.; data curation: S.A.W. and J.H.; writing and visualization: S.A.W. and J.H. Both authors have read and agreed to the published version of the manuscript.

**Funding:** This research received no external funding. Part of the work was completed while S.A.W. was employed by CognitiveScale.

**Institutional Review Board Statement:** Not applicable.

**Informed Consent Statement:** Not applicable.

**Data Availability Statement:** Datasets and code available at https://github.com/sinead/dmmd (accessed on 22 September 2021).

**Conflicts of Interest:** The authors declare no conflict of interest.

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
