# Peer review of "Understanding Collections of Related Datasets Using Dependent MMD Coresets"

_information, doi:10.3390/info12100392_

Round 1
Reviewer 1 Report
Accept with Minor Revisions
This paper proposes a method for MMD coresets for use in visualizing and interpreting similarity between pairs of distributions. MMD coresets represent discrete data via a kernel, effectively a kernel density estimate, and are a small weighted subset of the full data chosen carefully so that the L_infty error of the corresponding kernel density estimates is bounded.
The main idea of this paper is to restrict the small subset used to approximate the two distributions to be the same, but allow the weights to be different. The paper proposes a few algorithms for constructing such dependent coresets, and shows some visualization and data exploration examples where this aids in the understand.
The idea is not too complicated, but it is neat and worth while. It is one I wish I had. The examples demonstrate how it effectively provides a small summary of complex distributions that can be used to compare to elements.
I think it would be useful to discuss more the possibility of doing this not just for a pair (or small set) of objects to compare, but when you knew the comparison would come from a much larger set. Can this work as a natural extension of the proposed algorithms, or would this require substantial new ideas?
A few technical points:
The paper should cite the state-of-the-art results for the standard MMD coresets:
https://arxiv.org/abs/1802.01751
https://arxiv.org/abs/1906.04845
https://arxiv.org/abs/2007.08031
Note that the above results observe that random sampling provides a worst-case optimal coreset in some (very high-dimensional) settings, but also provides a potentially useful result in lower dimensions. As such, it may be better to temper some discussion about how just using a random sample has issues. Rather, a random sample does have (actually pretty strong) accuracy guarantees, but that in some (especially low-dimensional) settings, one can do substantially better.
Reviewer 2 Report
The study applied dependent MMD corsets to understand multiple related datasets and for models generalization. The study used an example of a photo images dataset of multiple decades to identify if models can classify the probability of each photo and where it belongs. The study separates datasets into two classes (1990s class and 2000s class). Similarly, the study used handwritten digits to classify images and to which of the two classes (early digits or late digits) they belong.
Did the two examples solve the problem, which should deal with unrelated datasets?
Also, how will the model behave if datasets have more than just two classes?
It was not clear in the text what problems this model will solve?
Other major issues:
- Authors should report more than the general accuracy as a performance measure (for example, recalls, precisions, false-positive rate, and G-scores, etc)
- Authors should add a section to discuss threats to the validity of the research work.
- The authors should elaborate more on the discussion part to see the model's usefulness and where it can be applied.
- Authors need to discuss the significance of the results and how they can solve real problems.
Round 2
Reviewer 2 Report
The authors have made significant changes to the original manuscript. The added sections improved the content and conveyed the contribution much better.
I still have some concerns about this paper:
1) Should the reported accuracy be a single number? Why X_a and X_b are reported separately? The confusion matrix shows the correct classification and misclassification of each class.
2) Are the classes of the images the 1940s, 1950s, 1960s.........2000s? If yes, why the confusion matrix was not reported for in this case?
3) Is there any explanation of the different performances of different algorithms? Why choosing these algorithms specifically?
4) I still cannot see threats to validity or limitation section included in the new version? This section is important to discuss the quality of the data, design of experiments issues, and generalizability
